# Short- and Long-Term Structural Characterization of Cured-in-Place Pipe Liner with Reinforced Glass Fiber Material

**DOI:** 10.3390/ijerph17062073

**Published:** 2020-03-20

**Authors:** Hyon Wook Ji, Dan Daehyun Koo, Jeong-Hee Kang

**Affiliations:** 1Korea Institute of Civil Engineering and Building Technology, Department of Land, Water and Environment Research, 283, Goyang-daero, Ilsanseo-gu, Goyang-si, Gyeonggi-do 10223, Korea; jihyonwook@kict.re.kr; 2Department of Engineering Technology, Indiana University-Purdue University Indianapolis (IUPUI), 799 W. Michigan St. ET 314J, Indianapolis, IN 46202, USA; dankoo@iupui.edu

**Keywords:** trenchless technology, glass fiber, flexural strength, flexural modulus, creep test, environmental engineering

## Abstract

Cured-in-place pipe (CIPP), as a kind of trenchless sewer rehabilitation technology, is a method to repair sewer pipe using unsaturated polyester resin. This study develops a CIPP liner using hot water or steam curing as well as glass fiber, in contrast to traditional methods, which use nonwoven fabric. Composite material samples were fabricated by combining liner materials using various methods, and the structural characteristics of the liners were compared and analyzed through short- and long-term flexural strength tests. A long-term test was conducted for 10,000 h, and the results revealed 13.3 times higher flexural strength and 8 times higher flexural modulus than the American Society for Testing Materials minimum criteria for CIPP short-term properties. The maximum creep retention factor was 0.64, thereby reducing the design thickness of the CIPP by up to 54%. The structural characteristics also improved when glass fibers were mixed with traditional CIPP liner, making it possible to reduce the thickness by 30%. Glass fibers result in high structural strength when combined with unsaturated polyester resin. Structural strength increased, even when glass fibers were mixed with traditional CIPP liner. The main contribution of this research is the development of a high strength CIPP liner and improvement of the structural properties of CIPP lining without using the specially formulated resin or lining materials.

## 1. Introduction

The purpose of sewer systems is to prevent urban flooding and maintain sanitary conditions. To this end, sewer systems rapidly remove and transport surface rainwater and deliver sewage to treatment plants. Sewer pipes are not visible because they are buried underground, but they are also gradually damaged due to continuous traffic, ground loads, groundwater effects, and aging. Aging sewer pipes crack, and the soil, tree roots, and groundwater that penetrate the cracks disrupt the flow in the pipes. Sewage leaking out of the cracks leads to internal and external problems, such as contaminating soil and causing sink holes [1]. To prevent or address these problems, repair or replacement is required.

Aging sewer pipes need to be replaced, but general open-cut excavations obstruct traffic and cause long-term inconvenience, thereby incurring social costs. Trenchless sewer rehabilitation, however, necessitates less traffic control [2], causes fewer complaints and environmental impacts [3], leads to lower accident risks [4], and improves productivity [5] and economic efficiency [6]. Therefore, it is desirable to extend the service lives of sewer pipes by rehabilitating them if possible.

Trenchless sewer rehabilitation includes various methods, such as pipe bursting, slip lining, and modified cross-section liners [7], but cured-in-place pipe (CIPP) has been most widely used. CIPP is a method in which a liner impregnated with thermosetting resin is placed into an existing sewer pipe, expanded by hydrostatic or air pressure, attached to the inner surface of the pipe, and cured on-site by applying heat [8,9]. The CIPP construction method can be largely divided into the manufacturing and site installation processes (Figure 1). In a factory, a liner is manufactured to fit the size of the existing sewer pipe. At the site, the condition of the pipe is inspected using closed-circuit television (CCTV). After cleaning the pipe, the liner is inserted, cured, cooled, and finished. When the construction is completed, only the cured liner is left inside the pipe for its structural rehabilitation. In terms of work hours, liner manufacturing takes a few days from material preparation to manufacturing, but a team can install and finish a length of approximately 100 m on-site in a single day. Although the field production rate may significantly vary depending on the site conditions, construction method, and workmanship, CIPP is a useful technology for urban sewer rehabilitation because it can significantly reduce traffic obstruction during the construction period and excavation, as compared to traditional excavation work.

A combination of various technologies is required from the start of CIPP to the completion of construction, but the key technology lies in manufacturing the CIPP liner itself. Traditional CIPP liner is made by impregnating nonwoven repair cloth with unsaturated polyester resin. As unsaturated polyester resin in the liquid state cannot maintain its shape, it is impregnated into fabrics, such as repair polyester felt, to maintain cylindrical shapes like those of sewer pipes. Unsaturated polyester resin is a thermosetting resin. It is made by adding diluents such as styrene to unsaturated polyester resin for free molding and by adding polymerization inhibitors to prevent polymerization during transportation. The resin hardens when an appropriate temperature is applied after adding an initiator and a catalyst [10]. Unsaturated polyester resin has high moldability, and is therefore used in various fields. The unsaturated polyester resin used in CIPP must have a high cure initiation temperature because it needs to stay outside for a long time, even in summer. CIPP resin must be sufficiently impregnated in the cylinder-shaped liner along the pipe wall, and the resin must be able to move inside the liner once it is installed inside the pipe. As the impregnation amount of the resin inside the liner must be maintained before and after curing, and handling must be easy, it is important for the resin to have an appropriate viscosity. In addition, conventional polyester resin for CIPP application needs to comply with the minimum requirement for the chemical resistance properties in standard domestic wastewater, as shown in Table 1. 

The economic efficiency of CIPP is determined by various elements, such as materials, tube manufacturing methods, and construction methods. Among these elements, the flexural structural property is the most important for determining the thickness of a CIPP liner. As the strength of CIPP increases, its design can be performed with a smaller thickness and fewer materials. A liner with a small amount of resin and a small thickness reduces on-site work hours due to the fast curing speed, and is economical because of the low cost for maintaining the heat required for curing decreases.

Traditional CIPPs usually have a large liner thickness because the minimum requirement for the flexural property is not high. Hot water or steam is used for curing, and a large amount of energy is consumed via the transportation of hot water and heating the liner to a curable temperature. On the other hand, CIPPs that use ultraviolet (UV) rays and glass fibers have high structural strength [12] and can save energy because they do not use hot water. In addition, the combination of epoxy and glass fibers [13], or epoxy and carbon fibers [14], can improve the flexural properties. However, epoxy and carbon fibers are relatively costly. Moreover, various resins have been developed and used, including vinyl ester resins with higher chemical resistance than polyester resin [15], and nonstyrene-based resin and silicate resin without styrene odor [13]. General unsaturated polyester resin is competitive in price compared to other resins, and thus, has been most widely used in CIPP liners for sewer rehabilitation worldwide.

The new trend of CIPP research and development has focused on environmental implications and high strength CIPP liners. Researchers have shown concern about styrene-based resin and its levels in drinking water sources after the application of a CIPP. Donaldson [16] revealed that several water sources after the application of CIPPs exceeded the Environmental Protection Agency’s maximum containment level for drinking water of 0.1 mg/L. Ra et al. [17] and other researchers reported that volatile and semivolatile organic compound (VOC/SVOC) emissions were detected during the sewer CIPP installation process. Environmental implications and other hazardous material emissions during the application of CIPP and their impact on humans are still the subject of debate, and need further research. Polyester resin manufactures developed nonstyrene resin systems for the application CIPPs, but the cost-benefit of using conventional resin systems has not been overcome.

CIPP liner manufacturers have introduced high strength liner materials. However, no similar academic research about the development of these materials, nor any details of their short- and long-term mechanical properties, has been published. Most previous research has focused on retrospective evaluations of existing CIPP liners for long-term quality control and assurance [18,19].

Hot water or steam curing, which is the traditional curing method, can transfer temperature and pressure evenly and in a stable manner, thereby enabling the construction of consistently good quality materials. In addition, this curing method does not require special curing equipment, except boilers, and exhibits excellent heat transfer characteristics. Therefore, the development of a high-strength liner that uses ordinary hot water and steam has the benefits of improving strength and quality and reducing costs.

In this study, the long- and short-term structural characteristics of a composite liner, made by adding glass fibers to unsaturated polyester resin and felt, were compared with those of a CIPP composite material liner that uses glass fibers for the development of an economical high-strength CIPP liner. To this end, composite material samples were fabricated by combining liner materials using various methods, and the structural characteristics of the liners were compared and analyzed through short- and long-term flexural strength tests.

## 2. Experimental Materials and Methods 

The CIPP liner samples used in this study were fabricated by combining the liner materials presented in Table 2 with isophthalic unsaturated polyester resin and through lamination and impregnation. A conventional unsaturated isophthalic polyester resin was diluted with styrene, providing low viscosity to facilitate resin impregnation through liner materials. Styrene was also used as a monomer participating in the free radical polymerization process. A small amount (approximately 1% of the resin weight) of peroxide was added to catalyze the free radicals into the polymerization process. The resin used in this study was produced by a mixture of isophthalic acid, unsaturated dicarboxylic acids, and glycols. Figure 2 shows a generic structure of isophthalic polyester resin, which is used for the majority of CIPP lining applications in the industry.

The basic thickness of the felt was 2 mm, but this was adjusted by stacking several layers when necessary. The polyester nonwoven felt (F) is polyester-based, as with unsaturated polyester resin. It has excellent material binding and adhesion performance, and the space of the felt structure induces and stores resin impregnation. The felt also secures sufficient tensile strength, thereby preventing excessive stretch or expansion/contraction and maintaining the shape of a CIPP liner during its installation inside a sewer pipe.

A polyester nonwoven felt with coating film (CF) was used on the outer surface of the liner. The impermeable film attached to the felt prevents damage to the tube and improves the liner curing quality by separating sewage from the resin. In addition, it protects the liner against various pollutants contained in the sewage, thereby ensuring appropriate curing and extending the service life of the liner (Figure 3a). The thickness of the felt and film was approximately 1.5 mm, and that of the film was 0.5 mm.

For the roving cross glass fiber and chopped strand mat (G), which is the main material of this study, one layer of 0.5 mm-thick glass fibers woven into a lattice was bonded to one layer of 0.5-mm-thick glass fibers that were finely chopped and attached like a nonwoven fabric. All the fibers were made of E-glass (Figure 3b). As this glass fiber had two layers, it was relatively thick. The handling was easy because the two layers maintained its shape, and it has various benefits, such as no directivity. The above materials were mixed using various methods, and tests were conducted to find the optimal structural characteristics.

The preparation of CIPP liner samples follows a standard practice used in the industry. The curing process used hot water as a heat source for the resin polymerization, but the potential degradation issue is eliminated by wrapping waterproof film layers. The water effect during the life cycle service of the CIPP liner is not considered, but the materials used in this study are not known to be degradable by aquatic conditions.

Samples for short- and long-term tests were fabricated in accordance with ASTM D790 [21]. The mold for the samples was a plate-type made of stainless steel with dimensions of 207 × 239 mm (Figure 4). The samples were prepared by placing the resin and the other materials into the mold and performing impregnation and curing. Each sample had different thicknesses, depending on the material combinations, but they were cut to the same length and width, i.e., 120 mm and 15 mm, respectively. A table saw was used for the cutting, and a diamond wheel blade for concrete, or a steel blade for acrylic resin, was used.

### 2.1. Short-Term Test

The liner of CIPP is a composite material that combines thermosetting plastics with reinforcing materials. The minimum strength criteria are a flexural strength of 31 MPa and a flexural modulus of 1724 MPa, as provided by the most widely used ASTM F1216 [11]. The domestic KS M 3550-9 [22] also specifies the same strength as the minimum value. These properties can be obtained by conducting the three-point bending test, suggested by ASTM D790, and using the following equations:(1)$$\sigma_{f}=\frac{3PL}{2bd^{2}}$$
and
(2)$$E_{B}=\frac{L^{3}m}{4bd^{3}},$$
where *σ_f_* is flexural stress (MPa). The maximum flexural stress before 5% flexural strain is the flexural strength (MPa). *P* is the load (N), *L* is the distance between points (mm), *b* is the width of the sample (mm), *d* is the thickness of the sample (mm), *E_B_* is the flexural modulus, and m is the initial slope of the load-strain curve (N/mm).

The flexural strength and modulus represent the magnitude of force-resisting bending compared to the thickness. In a three-point bending test, the load is increased until the bar-type sample, fabricated in accordance with the specifications suggested by ASTM D790 [21], ruptures, and the corresponding vertical displacement is measured.

### 2.2. Long-Term Test

The magnitude of the load that can be supported by a CIPP liner is determined by the thickness of the liner and the strength characteristics of the liner material. The thickness of the liner is determined using equations to obtain the load value for the liner thickness, which are given by ASTM F1216 [11] based on the theory on liner deformation characteristics under an external load. The equations are given below in Equation (3) for partial repair, which is used when the existing sewer pipe can sufficiently take care of external force, and Equation (4) for complete repair, which is used under the assumption that the existing sewer pipe can no longer maintain its structural performance. These equations originate from ASTM F1216 [11] expressions derived to obtain the liner thickness:(3)$$t=\frac{D}{\sqrt[{3}]{\frac{2KE_{L,50}C}{PN\left(1-v^{2}\right)}}+1},$$
and
(4)$$t=0.721D\sqrt[{3}]{\frac{{\left(\frac{Nq_{t}}{C}\right)}^{2}}{E_{L,50}R_{W}B\prime E_{s}\prime }},$$
where *t* is the thickness of the CIPP liner (mm), *D* is the inner diameter of the sewer pipe, *K* is the support improvement factor, *E_L,50_* is the 50-year long-term modulus of elasticity (MPa), *C* is the shape reduction factor, *P* is the groundwater pressure (MPa), *N* is the safety factor, *ν* is Poisson’s ratio, *q_t_* is the total external pressure applied to the pipe (MPa), *R_w_* is the buoyancy factor of water, *B’* is the elastic support coefficient, and *E’_s_* is the soil reaction coefficient (MPa).

*E_L,50_* is the long-term modulus of elasticity (*E_L_*) that remains 50 years after construction. *E_L_* is determined by the ratio of the flexural stress (Equation (1)) to the flexural strain when a certain load is given, as shown by: (5)$$E_{L}=\frac{\sigma_{f}}{r}.$$

In addition, the flexural strain is obtained using:(6)$$r=\frac{6D_{ef}d}{L^{2}},$$
where *r* is the flexural strain (mm/mm), and *D_ef_* is the deflection (mm). 

In the short-term test, the changing deflection was measured, and the flexural stress was calculated while the load was slowly changed. In the long-term test, however, the changing deflection was measured, and EL was calculated over time under the fixed load condition. The test methods followed the three-point bending test and sample specifications specified in ASTM D2990 [23] and ASTM D790 [21]. The weight of the long-term load used in the long-term test was calculated using Equation (7), as suggested by WIS 4-34-04 [24]:(7)$$M=\frac{bd^{2}S}{14.71L}\left(kg\right),$$
where *M* is the long-term load (kg), and *S* is the flexural strength (MPa), which is equal to 0.0025*E_B_*. *E_B_* is the flexural modulus. 

It is generally recommended that the long-term test be conducted for 10,000 h [25]. Based on the test data, the long-term modulus of elasticity at 50 years (438,000 h) is obtained through linear estimation on a log scale, and the value is described as *E_L,50_*. As 10,000 h exceeds one year, the test is sometimes replaced with 1,000 h test. In this study, however, the test was conducted for more than 10,000 h for a more standardized test. The deflection measurement intervals and frequencies were 30 times per minute, 9 times every 30 min, 45 times every hour, 45 times every 10 h, 20 times every 25 h, 80 times every 50 h, and 80 times every 80 h. The total measurement time was 11,400 h. The long-term modulus of elasticity was evaluated by calculating the creep retention factor (CRF, *C_L_*), which is the ratio of the long-term modulus of elasticity to the flexural modulus, shown in Equation (8), and comparing the values.
(8)$$C_{L}=\frac{E_{L,50}}{E_{B}}.$$

The typical values presented in the literature are 0.5 (polyester resin in nonwoven felt), 0.2 (epoxy in woven hose), and 0.7 (epoxy in fiberglass matrix) [11,13]. As *C_L_* is closer to 1, the material has better structural performance in the long-term because its deformation is small under a long-term load, making it possible to design a smaller thickness. 

## 3. Results and Discussion

### 3.1. Short-Term Structural Characterization

To test the short-term properties, five types of samples were fabricated through the combination of one unsaturated polyester resin for sewer repair and three reinforcing materials, as shown in Table 3. The sample F group was the control group and was fabricated using only the polyester nonwoven felt and resin. While F was fabricated for testing the properties of the traditional liner, the CFF group had the shape of a traditional CIPP tube that is installed in the field. The sample CFF group was fabricated by placing the polyester nonwoven felt with the coating film, on the top and bottom surfaces, the felt in the middle, and by finally impregnating them with the resin. The CF is expected to have no impact on the strength of the liner because it does not absorb the resin due to its impervious property, but there is no published report confirming this. This sample was fabricated to experimentally identify the effect of the CF on the CIPP liner.

The sample G group was fabricated by stacking roving across glass fiber, chopping the strand mats, and then impregnating them with the resin. This group is similar to the FRP structure that is used in various areas, such as ships, building materials, sporting goods, and bathtubs. It is strong and elastic, inexpensive, light, and highly resistant to environmental factors [26]. The CIPP liner cured by the UV method was installed after making a tube by wrapping the outer surface of these materials with impervious film, UV blocking film, and antidamage film. Sample G# (#: number of stacked layers) was fabricated from G4 to G7.

As for the FG sample group, the polyester nonwoven felt was placed on the top, and bottom surfaces and glass fibers were placed in the middle. Each layer was then impregnated with the resin. In general, CIPP liners that use glass fibers are combined with UV-curable resins, but this sample group was designed to use traditional hot water or steam curing. This group ranged from FG2F to FG5F, depending on the number of stacked glass fiber layers.

The sample CFG group has the outer surface wrapped with coating felt. It has a form in which glass fibers are added to the core of the traditional CIPP liner. CF was used to enable the actual installation of the FG group. Three glass fiber sheets were placed in the middle to increase strength.

The short-term property test results of these samples are shown in Figure 5. Figure 5a shows that the flexural strength of F was 48 MPa, and the corresponding flexural strain was 2.4%. The flexural strength of CFFCF, for which coating felt was added to F, was similar to that of F, but its strain was approximately double. Equation (1) shows that the flexural stress is inversely proportional to the square of the thickness. It was predicted that the impervious film contained by CFFCF would have no impact on strength because it does not absorb resin. Therefore, the flexural stress curve calculated, except the thickness of the coating felt (1 mm), was found to be CFFCF-C, which exhibited increased flexural strength but decreased strain compared to CFFCF. Compared to F, both the flexural strength and strain increased. These results indicate that impervious CF increases strain. As the increased flexural strain did not exceed 0.05 mm/mm (5%), the flexural strength was still effective.

Figure 5b shows the flexural properties of G, which was fabricated using only glass fibers and resin. Some of the experimental results of Ji et al. [27] were compared with those of F. The flexural strength of G was approximately nine times higher than that of F. In particular, a pattern whereby the curves smoothly increased and then dropped indicates that the glass fibers and the resin were completely integrated.

Figure 5c shows the curves of the FG group. Some of the experiment results of Ji et al. [27] were compared with those of F. The flexural strength was approximately three times higher than that of F, and the strain significantly increased. FG3F and FG4F were continuously resistant to the increasing load, even though the flexural strain exceeded 5%. For 5% or higher strain, however, the flexural strength is not effective in accordance with ASTM D790 [21].

Figure 5d shows the curve of the CFG group. There was no significant reduction in strength, even though three glass fiber sheets were inserted in the middle, and the coating felt was added. Due to the glass fibers, however, a failure occurred at a point where the flexural strain was approximately 5%. Here, the flexural stress and strain were also calculated, except the thickness of the coating felt (1 mm), and the results were found to be similar to CFG3CF-C, which exhibited increased flexural strength but decreased flexural strain. 

The short-term property test results of the samples are summarized in Table 4. An important factor in the design of CIPP is the flexural modulus. *E_L,50_* is proportional to *E_B_*, as shown in Equation (5), and the thickness of CIPP is inversely proportional to the cube of *E_L,50_*, as shown in Equations 3 and 4. If the flexural modulus increases, a thinner liner can be used.

The coating felt of CFFCF slightly increased the flexural strain and the flexural strength; however, it reduced the flexural modulus by more than 30%. The flexural modulus, except the coating felt, was similar to that of the F. Measures to improve strength while maintaining the constructability of the traditional CIPP are FGF and CFGCF. The flexural strength increased by approximately three times, and the flexural modulus increased by approximately 1.4 times compared to the original values.

When the structural characteristics of these materials were compared with the criterion of ASTM F1216 [11], CFFCF with the lowest flexural modulus exhibited an approximately 1.4-times higher value. G exhibited an approximately 8-times higher value, and FGF an approximately 2.8-times higher value. CFGCF exhibited an approximately 1.7-times higher value. According to NASTT [13], the flexural modulus and flexural strength of the CIPP liner made of nonwoven felt and polyester are 59 and 3,590 MPa, respectively. A comparison with these indicates that the addition of glass fibers contributes to strengthening structural characteristics, and CFGCF becomes a compromise plan that has the strength improvement effect while maintaining constructability. The presence of the impervious CF was found to reduce the flexural strength and the flexural modulus by approximately 30%.

### 3.2. Long-Term Structural Characterization

The test results showed that *E_B_* was high when only glass fibers were mixed with the unsaturated polyester resin. When the coating felt that improves constructability was added, however, an increase in the overall thickness of the liner was inevitable. The long-term load test was conducted on the following three samples: the samples that combined the unsaturated polyester resin with glass fibers (G4 and G5), which exhibited the highest *E_B_* in Table 4, and the sample that combined the resin, glass fibers, and coating felt (CFG3CF), which has excellent constructability.

For the test, two samples were prepared for each sample, and thus, a total of six samples were prepared. The long-term load (M) was calculated using Equation (7), and the results were found to be 10, 15, and 4.5 kg, respectively (Table 5). The corresponding weight was placed in the center of each sample, and a displacement meter was placed on top of it to measure the deflection (Figure 6). The measurement results were recorded at determined time intervals using a data logger.

In the long-term load test, the deflection measurement time is 10,000 h. Scatter plots and trend lines for x-y were drawn by converting the measurement time, which is the x-axis, into a log scale (Figure 7). Based on the trend lines, data are above the trend lines at the beginning of the measurement, but they are below the trend lines after 100 h. As the graphs are also on a log scale, the observation data after 100 h have the same importance as the data measured within the first ten hours, even though they were measured at considerable time intervals. In particular, the moment at which the weight was first loaded to each sample and the behavior of the next few minutes have a significant impact on the entire creep test.

*E_L,50_* can be obtained by substituting 438,000 h, corresponding to 50 years, into these three equations. G4 and G5, which have the same material and differ only in the number of glass fiber layers, have high slopes and y-intercepts. On the other hand, the slope and y-intercept of the trend line of CFG3CF exhibit low values. The trend lines of each case are given by: (9)y=−510.7ln(x)+13757,
(10)y=−1162ln(x)+23788,
and
(11)y=−219ln(x)+5182.2.

Table 5 summarizes the results of the long-term load test. G4 and G5 exhibited high *E_L,50_* values. G5 showed the highest CRF (0.64), followed by G4 (0.56). It can be said that the liners made of glass fibers have very high durability because their long-term, as well as short-term properties, are high. CFG3CF showed a relatively low CRF (0.51).

The minimum flexural modulus suggested by ASTM F1216 [11] is 1724 MPa. If the general CRF is 0.5, the long-term modulus of elasticity after at least 50 years is 862 MPa. Since G4 and G5 have 8.3- and 10-times higher values compared to this criterion, the liner thickness can be reduced by 51% and 54%. For CFG3CF, the liner thickness can be reduced by 30% because it has a 2.7-times higher value.

### 3.3. Discussion

The polyester felt used in liner composite materials increases the ductility of the liner, thereby increasing the flexural strain. In ASTM D790, the flexural strength represents the maximum flexural stress within 5% flexural strain. Thus, the flexural stress at 5% or higher strain is meaningless [21]. Depending on the samples, the felt increased the strain of the liner to more than 5%. On the other hand, glass fibers contributed to achieving effective flexural stress within the limited strain because they not only had the effect of increasing the flexural strength, but also decreased ductility and increased stiffness.

Glass fibers increase the strength and modulus of elasticity [28]. When the CRF values were compared, there was not a significant difference between the liner that used glass fibers and the one that used the felt together; however, there was a significant difference in the design thickness.

The thickness of a CIPP liner is adjusted by the 50-year modulus of elasticity. The change in the long-term modulus of elasticity is based on the deformation caused by the change in modulus of elasticity by using a test method for observing the strain through the long-term load test. The test to secure the quality of individual liners is practically very difficult because it requires a long time period, i.e., 10,000 h. Therefore, in the design method suggested by ASTM, the flexural modulus (a short-term property) is obtained. The long-term modulus of elasticity is obtained by multiplying it by the general CRF of 0.5, and is used for the CIPP thickness design. Using the existing CRF for new materials, however, sometimes plays a decisive role in underestimating good materials or overestimating bad ones. In NASTT [13], CRF varied from 0.2 to 0.7 depending on the combination of polyester, epoxy, felt, and glass fibers. Riahi [15] showed that CRF varied from 0.01 to 0.87, depending on the types of vinyl ester resins and glass fibers. To avoid uncertainty in material strength evaluation, the test on the long-term modulus of elasticity must be conducted.

The test, however, is highly likely to involve a high degree of uncertainty because it performs the extrapolation of the results after 50 years from the results obtained during a limited time period. Trenchless technology has expanded its application range in the sewer system market, and follow-up studies have been conducted on the existing CIPP [29]. These follow-up studies are significant because they provide insight into the long-term behavior of CIPP liners, and they can complement the long-term modulus of elasticity. It is necessary to establish more precise methodologies based on the results of these studies.

## 4. Conclusions

In this study, CIPP liners were fabricated in various combinations using glass fibers, which have been more frequently used in trenchless sewer rehabilitation, and the structural characteristics of each case were analyzed. When only glass fibers and unsaturated polyester resin were used, the flexural strength was 13.3 times higher, and the flexural modulus was 8 times higher compared to the minimum criteria of ASTM F1216 [11]. In addition, the flexural strength was 6.2 times higher, and the flexural modulus was 3.6 times higher than the general CIPP liner suggested by NASTT [13]. When the coating felt with impervious film was added to this to allow the use of traditional hot water and steam curing, the flexural strength decreased to 161 MPa and the flexural modulus to 4.59 GPa, but these values were still high compared to the existing liner.

The 50-year modulus of elasticity was approximately 8 GPa when only glass fibers and unsaturated polyester resin were used, and 2.3 GPa when the coating felt was added. The CRF, which represents flexural modulus degradation over time, ranged from 0.51 to 0.64. These values were slightly higher compared to the CRF (0.5) of the traditional liner that used the unsaturated polyester resin and felt suggested by NASTT [13]. The long-term modulus of elasticity has a direct effect on the design thickness of a CIPP liner, and it is possible to reduce the liner thickness by 30–54%.

Felt and coating felt increase ductility, which is lacking in resins, through curing with unsaturated polyester resin, but they reduce the flexural strength and the flexural modulus. As traditional CIPPs use only resin and felt, strength degradation cannot be prevented. Glass fibers, however, offset the strength degradation of felt by increasing the strength of a CIPP liner. The use of glass fibers with the composition of materials like that was suggested in this study can improve the strength of a liner while maintaining existing CIPP manufacturing and construction practices. 

## Figures and Tables

**Figure 1 ijerph-17-02073-f001:**
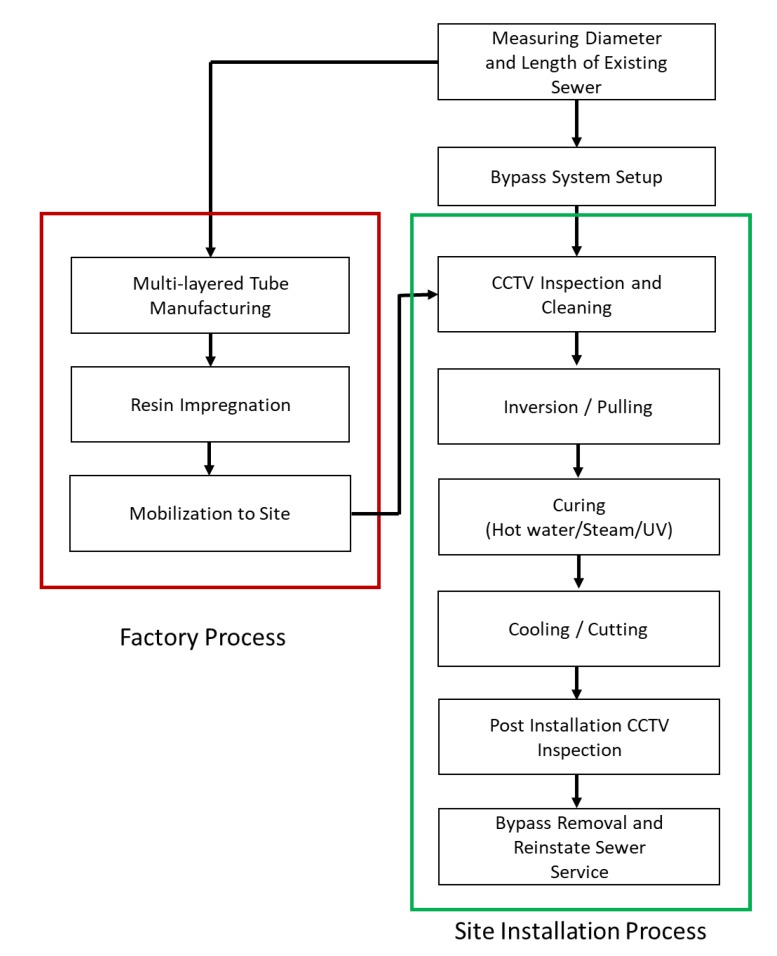
Cured-in-place pipe (CIPP) installation process.

**Figure 2 ijerph-17-02073-f002:**
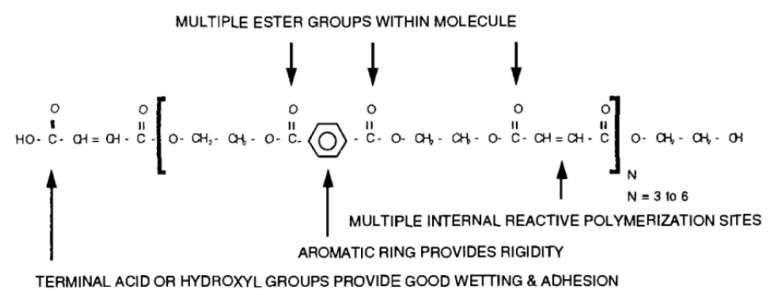
Generic structure of isophthalic polyester resin [20].

**Figure 3 ijerph-17-02073-f003:**
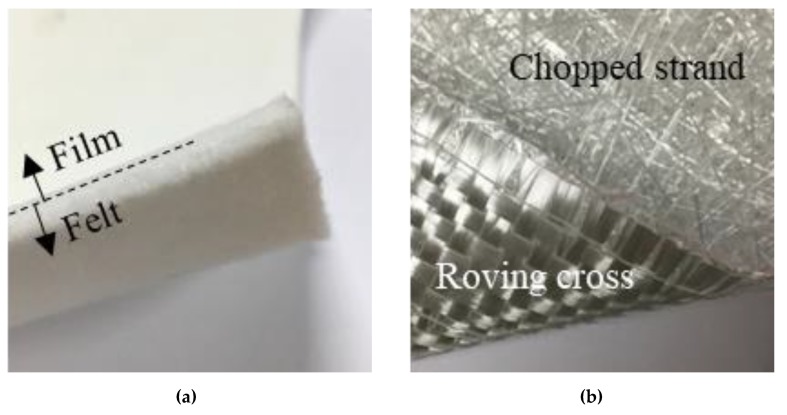
Appearance of materials for CIPP liner. (**a**) Polyester nonwoven felt with CF and (**b**) Roving cross glass fiber and chopped strand mat.

**Figure 4 ijerph-17-02073-f004:**
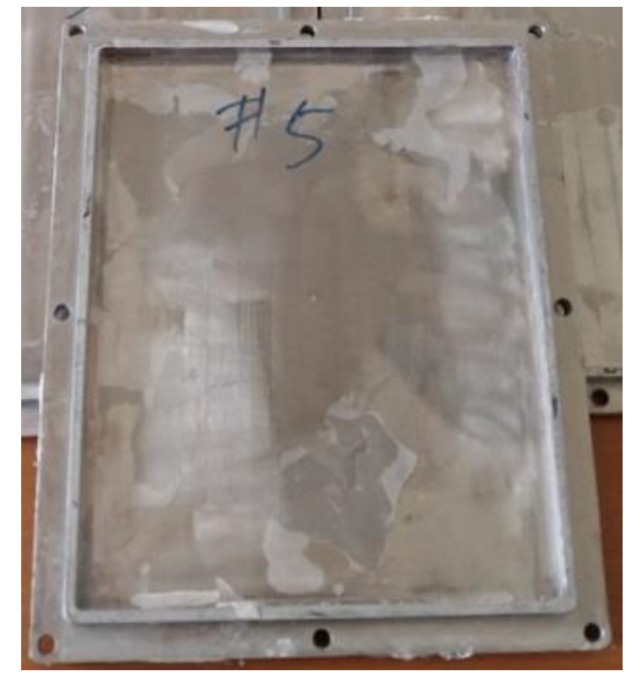
Mold for CIPP sample production.

**Figure 5 ijerph-17-02073-f005:**
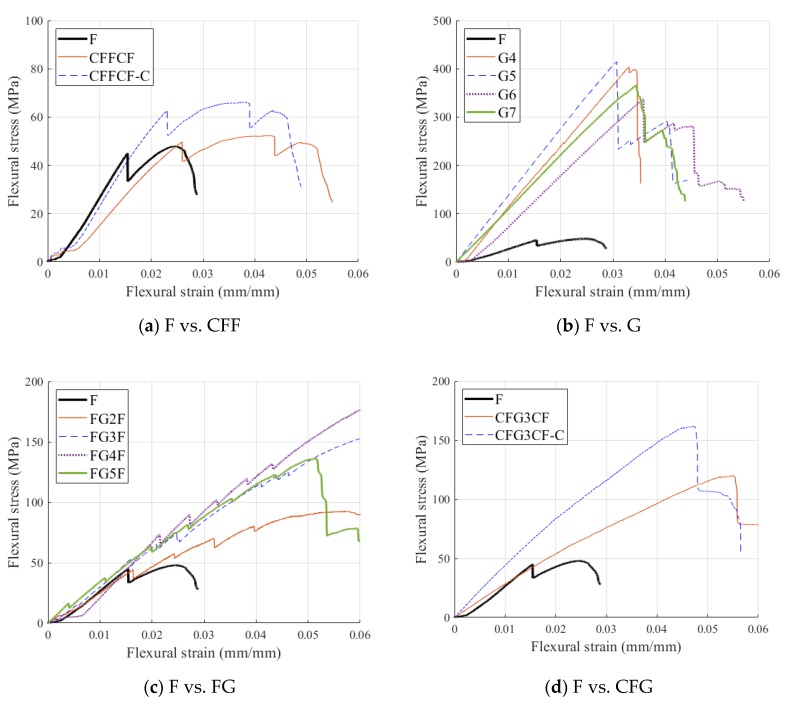
Flexural strain-stress curves of short-term property tests of CIPP liner samples: (**a**) F vs. CFF (**b**) F vs. G, (**c**) F vs. FG, and (**d**) F vs. CFG.

**Figure 6 ijerph-17-02073-f006:**
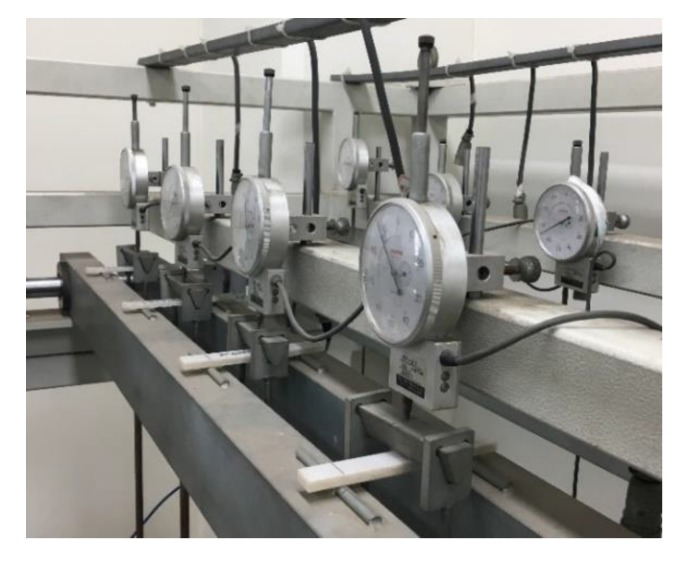
Creep test loading.

**Figure 7 ijerph-17-02073-f007:**
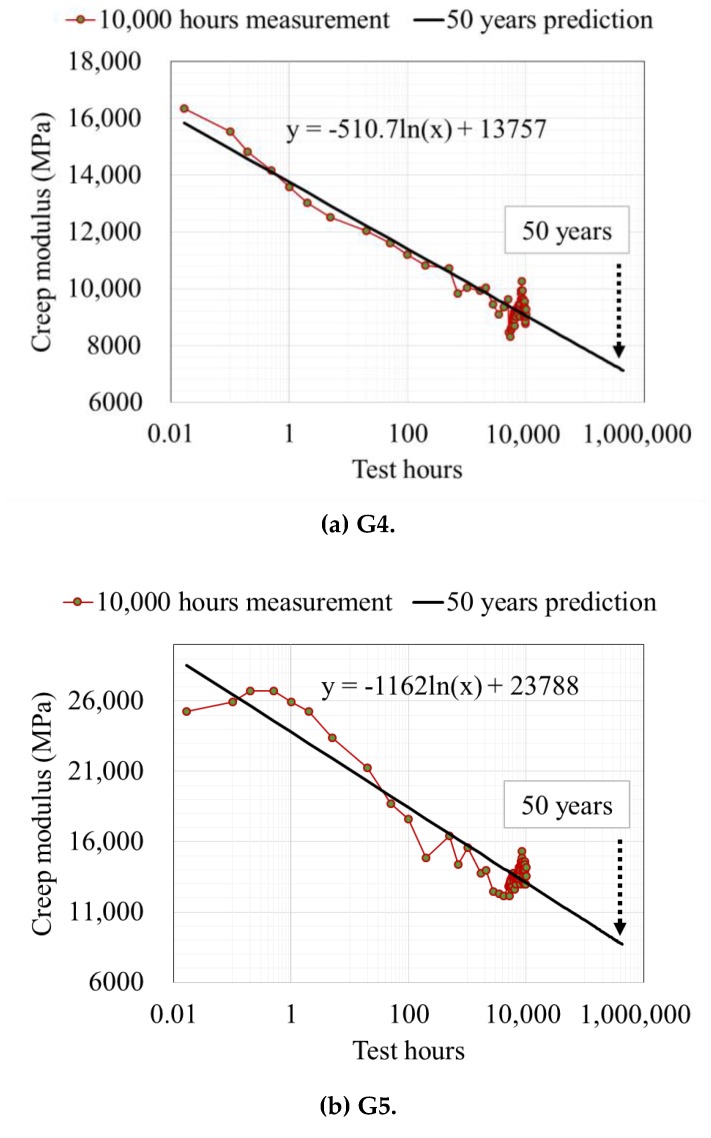
Experimental results of creep modulus for samples (**a**) G4, (**b**) G5, and (**c**) CFG3CF.

**Table 1 ijerph-17-02073-t001:** ASTM F1216 [11] minimum chemical resistance requirements for domestic sanitary sewer applications.

Chemical Solution	Concentration (%)
Tap water (pH 6–9)	100
Nitric acid	5
Phosphoric acid	10
Sulfuric acid	10
Gasoline	100
Vegetable oil	100
Detergent	0.1
Soap	0.1

**Table 2 ijerph-17-02073-t002:** Summary of materials and its characteristics for CIPP liner.

Material Name	Abbreviation	Thickness	Type
Unsaturated polyester resin	-	n/a (Liquid state)	Iso-type
Polyester nonwoven felt	F	2 mm	Polyester
Polyester nonwoven felt with coating film	CF	1.5 mm	Polyester
Roving cross glass fiber and chopped strand mat	G	1 mm	E-glass

**Table 3 ijerph-17-02073-t003:** Test samples.

Group	Samples Designations	Standard Images of Samples
F	F	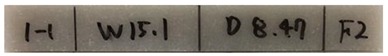
CFF	CFFCF	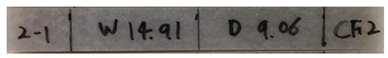
G	G4/G5/G6/G7	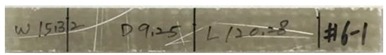
FG	FG2F/FG3F/FG4F/FG5F	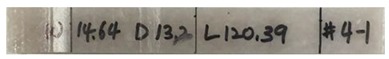
CFG	CFG3CF	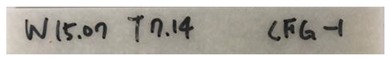

**Table 4 ijerph-17-02073-t004:** Short-term properties of CIPP samples.

Samples	Thickness (*d*, mm)	Flexural Properties		
		Strain (*r*, %)	Strength (*σ_f_*, MPa)	Modulus (*E_B_*, MPa)
F	8.47	2.4	48	3488
CFFCF	9.06	4.3	52	2309
CFFCF excluding C	8.06	3.8	66	3279
G4	5.55	3.3	403	13,800
G5	5.85	3.1	414	13,700
G6	8.86	3.6	337	10,490
G7	9.03	3.4	365	10,820
FG2F	6.61	5	90	3474
FG3F	7.64	5	132	4718
FG4F	9.44	5	150	4830
FG5F	13.16	5	135	4200
CFG3CF	7.14	5	115	2917
CFG3CF excluding C	6.14	4.7	161	4586

**Table 5 ijerph-17-02073-t005:** Mechanical properties with long-term modulus of samples.

Samples	*d* (mm)	*E_B_* (MPa)	*M* (kg)	*E_L50_* (MPa)	*C_L_* (CRF)
G4	5.14	12,800	10	7121	0.56
G5	5.94	13,600	15	8688	0.64
CFG3CF excluding C	5.96	4590	4.5	2336	0.51

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
