# Peer review of "Short- and Long-Term Structural Characterization of Cured-in-Place Pipe Liner with Reinforced Glass Fiber Material"

_ijerph, 2020, doi:10.3390/ijerph17062073_

Round 1
Reviewer 1 Report
- This study has developed a new CIPP liner using hot water and steam curing methods. It has also performed short- and long-term flexural strength tests to compare and analyze structural characteristics of the liner. The paper is well written and clearly summarized methodology and results of the study. I believe this research contributes trenchless technology by developing the better and secure structural CIPP liner.
- The authors provide good and concise abstract. It would be great if they can mention contribution of this study in one or two sentences in the abstract.
- Line 109: Should this sub heading be located in the next page?
- Would the section 3.1 be more appropriate as “Short-term Structural Characterization”? or “Short-term Flexural Strength Test (Based on the title of the paper?)
- Would the section 3.2 be more appropriate as “Long-term Structural Characterization?” (Based on the title of the paper?)
- Table 1: => better with “Summary (or Samples) of Materials and Its Characteristics for CIPP Liner”??
- Please double-check the font size and style of figure 6.
- Line 385-387: The statement is a little bit confusing and unclear. Please revise the sentence.
Reviewer 2 Report
The reviewed article concern structural characterization of cured-in-place-pipe liner with reinforced glass fiber material. Undoubtedly, the article contains a practical aspect, but I have doubts about the scientific novelty of the presented research. The form of the article presents a general research report rather than a scientific article. In part theoretical, there is no in-depth description of the current state of knowledge on the topic presented and indications of scientific novelty of presented research.
There is no detailed description of the reagents used, if they were commercial products, their name should be given. The statement "unsaturated polyester resin" - says little about a specific chemical structure (please provide formulas).
The authors focused on mechanical tests of the composites obtained, but were the tests performed in the aqua environment? In such conditions, such materials will, after all, work.
Due to the high IF journals, authors must provide information on the compounds used, because only then these studies can be of interest and useful to other scientists.
In addition, please develop what type of corrosion the authors mean: 77-78 ... "resin must satisfy various criteria such as high resistance to corrosion caused by various pollutants".
Round 2
Reviewer 2 Report
The article can be published in its current form.